# Nanoparticle display of prefusion coronavirus spike elicits S1-focused cross-reactive antibody response against diverse coronavirus subgenera

Geoffrey B. Hutchinson[1,2,3,13], Olubukola M. Abiona[1,4,13], Cynthia T. Ziwawo[1], Anne P. Werner[1,5], Daniel Ellis[2,6], Yaroslav Tsybovsky[7], Sarah R. Leist[8], Charis Palandjian[9], Ande West[8], Ethan J. Fritch[10], Nianshuang Wang[11], Daniel Wrapp[11], Seyhan Boyoglu-Barnum[1], George Ueda[2,6], David Baker[2,6,12], Masaru Kanekiyo[1], Jason S. McLellan[11], Ralph S. Baric[8,10], Neil P. King[2,6], Barney S. Graham[1] ✉ & Kizzmekia S. Corbett-Helaire[1,9] ✉

Multivalent antigen display is a fast-growing area of interest toward broadly protective vaccines. Current nanoparticle-based vaccine candidates demonstrate the ability to confer antibody-mediated immunity against divergent strains of notably mutable viruses. In coronaviruses, this work is predominantly aimed at targeting conserved epitopes of the receptor binding domain. However, targeting conserved non-RBD epitopes could limit the potential for antigenic escape. To explore new potential targets, we engineered protein nanoparticles displaying coronavirus prefusion-stabilized spike (CoV_S-2P) trimers derived from MERS-CoV, SARS-CoV-1, SARS-CoV-2, hCoV-HKU1, and hCoV-OC43 and assessed their immunogenicity in female mice. Monotypic SARS-1 nanoparticles elicit cross-neutralizing antibodies against MERS-CoV and protect against MERS-CoV challenge. MERS and SARS nanoparticles elicit S1-focused antibodies, revealing a conserved site on the S N-terminal domain. Moreover, mosaic nanoparticles co-displaying distinct CoV_S-2P trimers elicit antibody responses to distant cross-group antigens and protect male and female mice against MERS-CoV challenge. Our findings will inform further efforts toward the development of pan-coronavirus vaccines.

Coronaviruses (CoVs) comprise a family of viruses with diverse strains known to infect mammals and birds. Betacoronaviruses (β-CoVs), in particular, thrive in animal reservoirs and represent a constant threat to human health. Inclusive of the most recently emerged β-CoV, SARS-CoV-2, there are seven CoVs known to infect humans (human CoVs or hCoVs); four of which circulate endemically[1,2]. Although endemic hCoV infections typically manifest as mild respiratory disease, zoonotic spillover of β-CoVs into human populations has been associated with

high disease morbidity and mortality, economic burden, and widespread global epidemics[3]. Since the emergence of SARS-CoV-2 in late 2019, millions of deaths worldwide[4] have been attributed to COVID-19[2,5] and as evidenced by ongoing circulation, SARS-CoV-2 will continue to evolve. The emergence of SARS-CoV-2 variants of concern (VOC) highlights the threat of viral escape from antibodies induced by the currently available COVID-19 vaccines and the need for next-generation vaccines that are capable of inducing broadly protective

immunity against a wide range of CoVs[6]. Recently, there has been documentation of broad cross-reactivity either deriving from natural infection[7] or from immunogens delivering multiple sarbecoviral antigens[8–10]. There is much interest in designing immunogens to target antibody responses to domains of the S1 subunit at the apex of the spike (S) protein[11,12]—particularly the receptor binding domain (RBD), as this region is targeted by potently neutralizing antibodies (nAbs)[13]. However, as evidenced by the omicron VOC, the RBD is particularly susceptible to mutation and recombination, leaving room for immune evasion via antigenic drift and shift. Others have previously shown elicitation of broad protection against influenza through the multi-valent display of hemagglutinin on the I53_dn5 nanoparticle platform[14]. Applying lessons learned from those studies, here we describe nano-particles that display S trimers from diverse coronaviruses. We show that CoV prefusion-stabilized S (CoV_S-2P) trimers displayed on I53_dn5 self-assembling icosahedral nanoparticles are able to elicit broadly cross-reactive and protective antibody responses. Further-more, by co-displaying these diverse spikes in a mosaic antigen array, we induce robust and protective immunity even at low valency of individual S proteins.

## Results

### Immunogen design and characterization

The introduction of two proline mutations (2P) at the apex of the central helix of a wide range of coronavirus spike proteins has pre-viously been shown to stabilize the prefusion conformation and elicit potent antibody responses[15–17]. We applied these stabilizing mutations to the S proteins of MERS-CoV, SARS-CoV-1, and SARS-CoV-2 Wuhan-Hu-1, the three epidemic and pandemic-causing β-hCoVs, and adapted these antigens for display on the computationally designed two-component nanoparticle, I53_dn5[18] (Fig. 1a). S-2P antigens were genetically fused to the trimeric component I53_dn5B and assembled in vitro by the addition of the pentameric component I53_dn5A to generate monotypic particles displaying 20 copies of the specified S-2P trimer[18]. Size exclusion chromatography (SEC) revealed peaks corresponding to S-2P_I53_dn5 nanoparticles indicating efficient assembly and formation (Fig. 1b). Purified S-2P trimers and nano-particles were tested for antigenicity by ELISA using monoclonal

antibodies specific to each CoV S. Antibody binding was comparable between soluble trimer and nanoparticle in each case indicating that antigenicity is similarly intact in each formulation (Fig. 1c). The purified nanoparticles were also imaged by negative stain electron microscopy (NS-EM) showing that the nanoparticles were well-assembled and homogeneous, displaying highly-ordered S proteins (Fig. 1d). These characterization data show that we were able to efficiently express and assemble antigenically intact nanoparticle immunogens.

### Display of SARS-1_S-2P on I53_dn5 qualitatively alters antibody cross-reactivity and potency

We hypothesized that multivalent display of CoV S-2P trimers would elicit more robust antibody responses to conserved and/or sub-dominant epitopes consequently improving potency and breadth of cross-reactivity when compared to soluble S-2P. To test this, we immunized BALBc/J mice twice with 10 μg of SARS-1_S-2P as a soluble trimer fused to the foldon trimerization domain or displayed on I53_dn5 nanoparticles, as well as MERS_S-2P or influenza HA (H1) displayed on I53_dn5. Immunizations were performed with Sigma adjuvant system (SAS), an oil-in-water emulsion derived from com-ponents of the bacterial cell wall, which is comparable to Freund's adjuvant. Mice were immunized at weeks 0 and 3 and then assessed for serological responses at week 5. Immunization with SARS-1_S-2P induced similar antibody binding titers against SARS-1_S-2P, SARS-2_S-2P, and MERS_S-2P as those elicited by SARS-1_I53_dn5 (Fig. 2a–c). Both SARS-1_S-2P and SARS-1_I53_dn5 elicited comparable strain-matched neutralizing antibodies against SARS-1-CoV pseudotyped virus (Fig. 2d), but neither neutralized SARS-2-CoV (Fig. 2e). Inter-estingly, SARS-1_I53_dn5-elicited antibodies were able to neutralize the MERS pseudotyped virus at levels similar to MER-S_I53_dn5 (Fig. 2f).

To assess the quality of the vaccine-matched and cross-reactive antibody responses, we generated correlation plots of neutralizing-to-binding antibody titers for SARS-1, SARS-2, and MERS, in which the slope indicates the ratio of neutralizing to binding activity. Because SARS-1_S-2P-elicited antibodies binding SARS-1_S-2P exceeded the limit of detection, a reliable SARS-1 correlation plot could not be generated; however, particle assembly reduced SARS-1_S-2P binding titers while

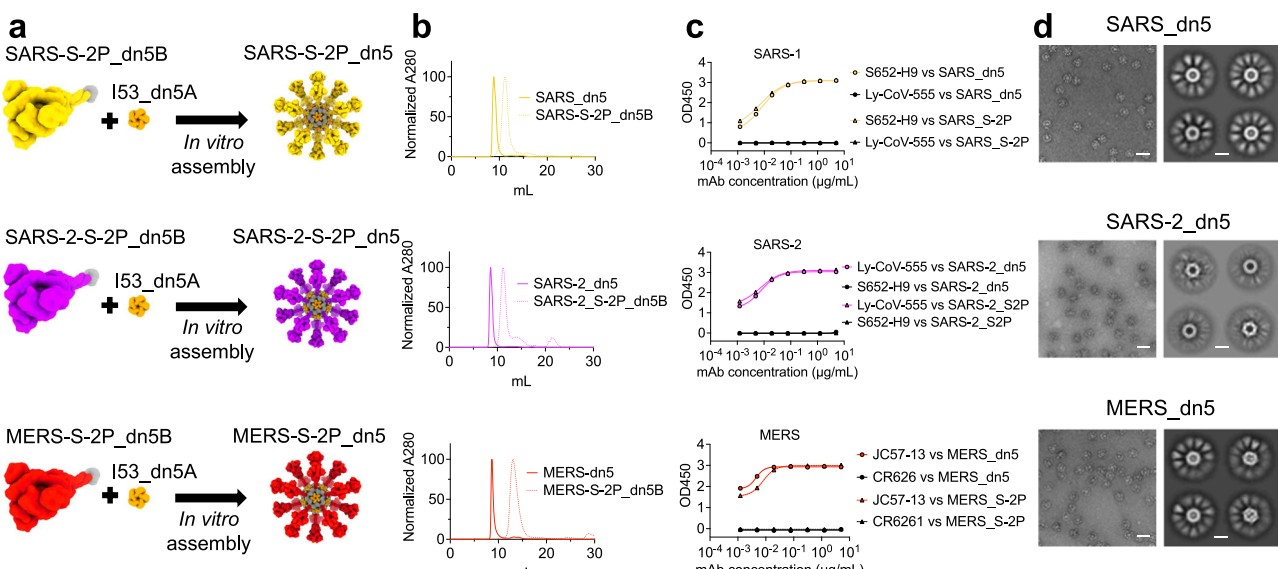

**Fig. 1 | Design and characterization of CoV-S-2P displayed on I53-dn5.**
**a** Computer-generated models of prefusion-stabilized spike trimers (S-2P) from SARS-1, SARS-2, and MERS, and their homotypic display on the icosahedral I53-dn5 nanoparticle displaying 20 trimers. **b** Trace profiles of S-2P_dn5B trimer and _dn5 nanocage purification by size exclusion chromatography. **c** ELISA comparing

binding of antibodies specific for SARS-1, SARS-2, or MERS_S-2P to soluble trimer (triangles) or dn5 assembly (circles) respectively. **d** Representative images of CoV-S-2P_dn5 at 50,000× magnification and 2D class averages. Scale bars correspond to 100 nm (representative images) and 20 nm (2D class averages). All experiments were performed at least twice, each repeat with similar results.

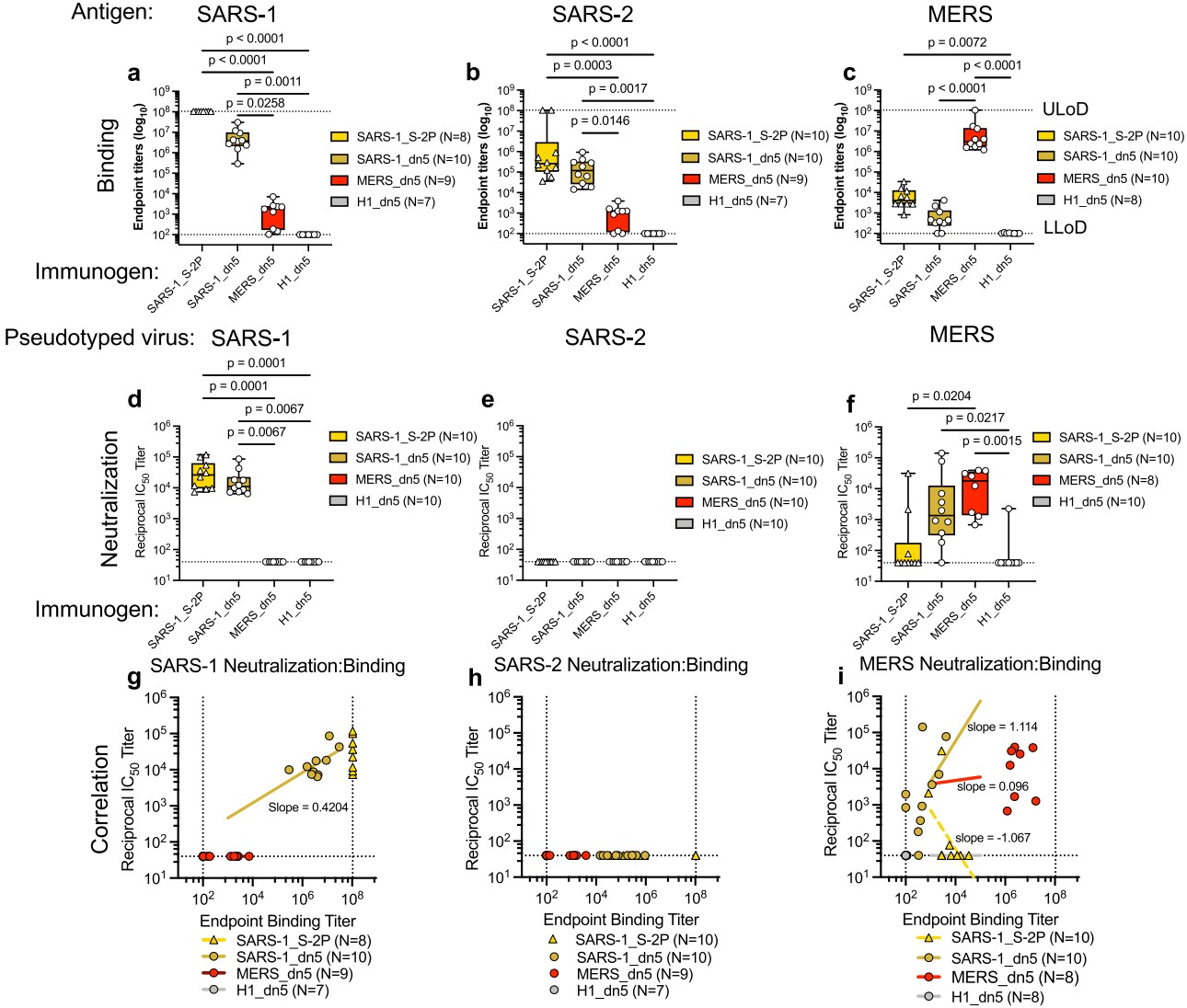

**Fig. 2 | Assembly of SARS-1_S-2P on dn5 elicits potent cross-neutralizing antibodies. a–f** Groups of 10 female BALB/cJ were immunized at weeks 0 and 3 with 10 µg of SARS-1_S-2P as a soluble trimer or displayed on I53_dn5 particles, MERS_S-2P trimer or H1 trimer displayed on I53_dn5 nanoparticles with SAS adjuvant and bled at week 5 for serology. Control mice were immunized with H1_dn5. **a–c** Sera were screened for binding by ELISA to SARS-1_, SARS-2_, and MERS_S-2P. **d–f** Serum was then assessed for its capacity to neutralize SARS-1, SARS-2, and MERS pseudotyped viruses. **g–i** To plot the potency of neutralizing antibodies, correlation plots of binding (x-axis) to neutralization (y-axis) where the slope (neutralization/binding) indicates the ratio of neutralizing to binding antibody titers were generated. **a–f** Boxes and horizontal bars denote the interquartile range (IQR) and medians, respectively. Whisker endpoints are equal to the minimum and maximum values. Statistical analysis was performed using the nonparametric Kruskal–Wallis test with Dunn's multiple comparisons. *$P < 0.05$, **$P < 0.01$, ****$P < 0.0001$.

neutralization remained intact. This suggests that particle assembly improved SARS-1 neutralization potency (Fig. 2g). SARS-2 correlation plots could not be generated due to the lack of detectable neutralizing antibodies (Fig. 2h). MERS correlation plots comparing the quality of MERS-binding and neutralizing antibodies elicited by SARS-1_S-2P, SARS-1_I53_dn5 and MERS_I53_dn5 indicate that while the overall magnitude of MERS-neutralizing and binding antibody titers elicited from MERS_I53_dn5 nanoparticles was higher, SARS-1_I53_dn5 nanoparticles elicited more potently neutralizing antibodies with higher neutralization-to-binding ratios (Fig. 2i). To further validate the SARS-1_I53_dn5 immunogen, we performed an ELISA with SARS-specific (S652-27) and MERS-specific (JC57-13) monoclonal antibodies (mAbs). S652-27 bound SARS-1_I53_dn5, but neither JC57-13 nor influenza-HA-specific mAb CR6261 bound, indicating that SARS-1_I53_dn5 immunogens were free of MERS spike contamination (Supplementary Fig. 1).

We also performed a dose escalation study in which C57BL/6 mice were immunized at weeks 0 and 3 with 0.1, 1.0, or 10 µg of SARS-1_S-2P

or SARS-1_I53_dn5. ELISAs revealed that SARS-1_I53_dn5-elicited IgG antibodies able to bind MERS_I53_dn5 were dose-dependent and only evident at the 10 µg dose (Supplementary Fig. 2). Sera from the 10 µg dose of this study as well as from 2 to 3 independent studies also using C57BL/6 mice immunized with 10 µg of SARS-1_S-2P, SARS-1_I53_dn5, or MERS_I53_dn5 were assessed for binding and neutralization. Aggregated data were used to generate neutralization-to-binding correlation plots (Supplementary Fig. 3). Notably, in C57BL/6 mice, I53_dn5 assembly of SARS-1_S-2P reduced cross-binding to SARS-2_S-2P while binding to MERS-S-2P increased (Supplementary Fig. 3a, b). MERS cross-neutralization from SARS-1_I53_dn5 nanoparticles was also found in C57BL/6 mice (Supplementary Fig. 3f). Soluble SARS-1_S-2P and SARS-1_I53_dn5 nanoparticles elicited similar quality antibodies against SARS-1, as indicated by their slopes, and neither group elicited neutralizing antibodies against SARS-2 resulting in a horizontal slope (Supplementary Fig. 3g, h). MERS correlation plots generated from C57BL/6 mice also show higher overall magnitude of binding and

neutralizing antibody responses elicited by MERS_I53_dn5 compared to SARS-1_I53_dn5, but comparable correlations of neutralization-to-binding (Supplementary Fig. 3i). Together, these data indicate that while SARS-1_I53_dn5 elicited MERS-reactive antibodies that were relatively low-binding, they were still able to potently neutralize MERS-CoV pseudotyped virus at a similar or greater level than MERS_I53_dn5 in two genetically distinct mouse strains.

## Cross-reactive antibodies target the S1 domain of spike

We hypothesized that an antibody derived from a rare B cell population specific for a conserved site of vulnerability on the MERS-spike could be responsible for the potent cross-neutralization despite lower binding capacity. To understand the domain specificity of cross-reactive antibodies elicited by SARS_I53_dn5 to MERS_S-2P, we performed S domain-specific depletion assays where sera from mice immunized with SARS_I53_dn5 were depleted with either the full-length MERS_S-2P trimer or its subdomains: S1, S2 (stabilized stem, SS)[19], or RBD. Depleted sera were then tested for residual binding to SARS-1_S-2P, SARS-2_S-2P, and MERS_S-2P. Depletion with MERS_S-2P or _S1 effectively eliminated binding to SARS-2_S-2P and MERS_S-2P, while binding remained intact after depletion with MERS_S2 (Fig. 3b, c). This indicates that cross-reactive antibodies induced by immunization with SARS_I53_dn5 to SARS-2_S-2P and MERS_S-2P are specific to the S1 subunit. Notably, depletion with MERS RBD did not result in a statistically significant loss in SARS-2 or MERS-binding antibody titers (Fig. 3b, c), suggesting that there are potent and highly conserved S1 non-RBD epitopes, such as in the N-terminal domain (NTD). Interestingly, when sera depleted of MERS-specific antibodies were assayed for binding to HKU1_S-2P, binding patterns were the same as for non-depleted sera; MERS_S1 depletion did not reduce binding to HKU1_S-2P as it did against SARS-2_S-2P and MERS_S-2P, indicating that the shared epitopes responsible for cross-reactivity elicited by SARS-1_I53_dn5 against SARS-2_S-2P and MERS_S-2P are distinct from the cross-reactive epitopes between SARS-1_S-2P and HKU1_S-2P (Supplementary Fig. 4).

We hypothesized that after nanoparticle assembly, crowding of the CoV S antigens on the nanoparticle surface sterically restricts B cell receptor (BCR) access to the S2 subdomain and focuses antibody responses to S1. To investigate this in the context of MERS S, we immunized mice once with 10 μg of either MERS_S-2P, MERS_I53_dn5, or bare I53_dn5 and tested for antibody binding to MERS_S-2P and stabilized MERS S2 domain (MERS_SS) at week 3. Both soluble MERS_S-2P and MERS_I53_dn5 nanoparticles elicited antibodies that bound to MERS_S-2P (Supplementary Fig. 5a). Interestingly, while MERS_I53_dn5 elicited greater binding antibody titers to full-length MERS_S-2P), only the soluble MERS_S-2P immunogen was able to elicit binding antibodies to the MERS S2 domain (Supplementary Fig. 5), suggesting that access to S2 is restricted on I53_dn5 and may bias toward selection of B cells specific to the S1 domain. To see if antibodies cross-reactive to S1 could be visualized by EM, sera from mice immunized with SARS-1_I53_dn5 were pooled; IgG was isolated and digested with papain to generate antigen-binding fragments (Fabs). We assembled immunocomplexes of SARS-1_I53_dn5-elicited polyclonal Fabs bound to MERS_S-2P and imaged them with negative stain EM. Both representative images and 2D class averages show Fabs bound to the apex and side of the MERS-S1 (Fig. 3d, e). We next generated an NS-EM 3D reconstruction of the complex and overlayed it with the structure of MERS_S-2P (PDB: 5X5F). The apex-binding Fab appeared to target a similar epitope as the previously characterized MERS-specific, NTD-binding mAb, G2 (PDB: 6PXH), while only a partial density could be generated for the side-binding Fab (Fig. 3f, g)[20]. Together, these data indicate that the assembly of SARS-1_S-2P on I53_dn5 restricts BCR access to the S2 domain and focuses B cell selection and antibody responses to S1. The potential avidity bonus from the multivalent display may provide an advantage to unique, potentially low-affinity, yet cross-reactive B cells specific for vulnerable sites on S1.

## Display of diverse CoV spikes on I53_dn5 elicits cross-reactive antibodies

We next sought to profile the breadth of cross-reactive antibody responses elicited from additional β-CoV S-2P antigens displayed on I53_dn5 as well as a mosaic nanoparticle assembled to co-display each of the β-CoV_S-2P antigens on I53_dn5. HKU1_ and OC43_I53_dn5 nanoparticles were generated as above (Supplementary Fig. 6) and BALB/cJ mice were immunized at weeks 0 and 3 with 10 μg of either SARS-1_, SARS-2_, MERS_, HKU1_, OC43_, or β-CoV mosaic on I53_dn5 nanoparticles and bled to assess antibody binding to the S-2P proteins derived from SARS-1, SARS-2, MERS, HKU1, OC43, as well as two α-hCoVs, hCoV-229E and hCoV-NL63 (Fig. 4a). H1_I53_dn5-immunized mice served as negative controls. SARS-1_I53_dn5 nanoparticles elicited cross-reactive antibody responses to MERS_S-2P and SARS-2_S-2P; however, binding was also detected against HKU1_S-2P and OC43_S-2P. SARS-2_I53_dn5 particles also resulted in broad binding, extending to SARS-1_, MERS_, HKU1_, and OC43_S-2P. MERS_I53_dn5 particles elicited cross-binding antibodies against SARS-1_, SARS-2_, HKU1_, and OC43_S-2P. HKU1_I53_dn5 elicited cross-binding to MERS_, SARS-1_, SARS-2_, and OC43_S-2P (Fig. 4b–f). OC43_I53_dn5 elicited cross-reactive antibodies to SARS-1_, SARS-2_, MERS_S-2P, HKU1_, and even α-CoV-derived 229E_S-2P (Fig. 4b–g). We also immunized C57BL/6 mice with these groups and found similarly broad, yet lower magnitude cross-reactivity (Supplementary Fig. 7). Mice immunized with a β-CoV mosaic nanoparticle co-displaying MERS_, SARS-1_, SARS-2_, HKU1_, and OC43_S-2P showed comparably high antibody responses to their respective corresponding monotypic I53_dn5 nanoparticle (Fig. 4b–f). Notably, the β-CoV mosaic nanoparticle also elicited cross-binding antibodies to 229E_S-2P (Fig. 4g), indicating that despite the reduced valency of OC43_S-2P per particle, the mosaic was able to retain its contribution of cross-reactive breadth[14].

## SARS-1_ and β-CoV mosaic_I53_dn5 assemblies protect against MERS-CoV challenge

To determine the protective efficacy of CoV S-2P immunogens using the I53_dn5 platform, we immunized 288/330[+/+] mice twice with 10 μg of either MERS_I53_dn5, SARS-1_I53_dn5, SARS-2_I53_dn5 or the β-CoV mosaic_I53_dn5 nanoparticle, using the influenza H1_I53_dn5 nanoparticle as a control[21]. Additionally, we immunized mice with SARS-1_S-2P or SARS-2_S-2P soluble protein to compare protection between soluble protein and I53_dn5-based immunogens. Mice were then subjected to a lethal challenge with a mouse-adapted MERS-CoV strain, MERS-CoV m35c4. SARS-1_S-2P and SARS-2_S-2P, which induced sub-neutralizing cross-reactive antibodies against MERS_S-2P, failed to protect against weight loss or lung disease following challenge (Fig. 5). However, the SARS-1_I53_dn5 nanoparticle induced cross-subgroup protection at a level approaching that of MERS_I53_dn5, with both groups sustaining body weight throughout the course of the challenge (Fig. 5a). These findings are consistent with our serology data showing that SARS-1_I53_dn5 elicits MERS-binding and neutralizing antibodies that correlate similarly to those elicited by MERS_I53_dn5 (Supplementary Fig. 3b, f, i). Additionally, for SARS-1_I53_dn5 and SARS-2_I53_dn5, lung discoloration scores were lower relative to their S-2P immunized counterparts (Fig. 5b), though viral loads were only marginally lower for SARS-1_I53_dn5 compared to SARS-1_S-2P (Fig. 5c). Nonetheless, these trends indicate that display of heterotypic SARS-1_ and SARS-2_S-2P on the I53_dn5 platform augments baseline immunogencity and cross-reactivity to MERS-CoV. β-CoV mosaic nanoparticles with decreased MERS_S-2P valency elicited protective responses to a similar degree as MERS_I53_dn5, as discerned by weight loss (Fig. 5a). In addition, mice immunized with the β-CoV mosaic nanoparticles had low lung discoloration scores (Fig. 5b) and lung viral titers (Fig. 5c), similar to mice immunized with MERS_I53_dn5 nanoparticles, again indicating that mosaic nanoparticles with a lower MERS_S-2P molar content

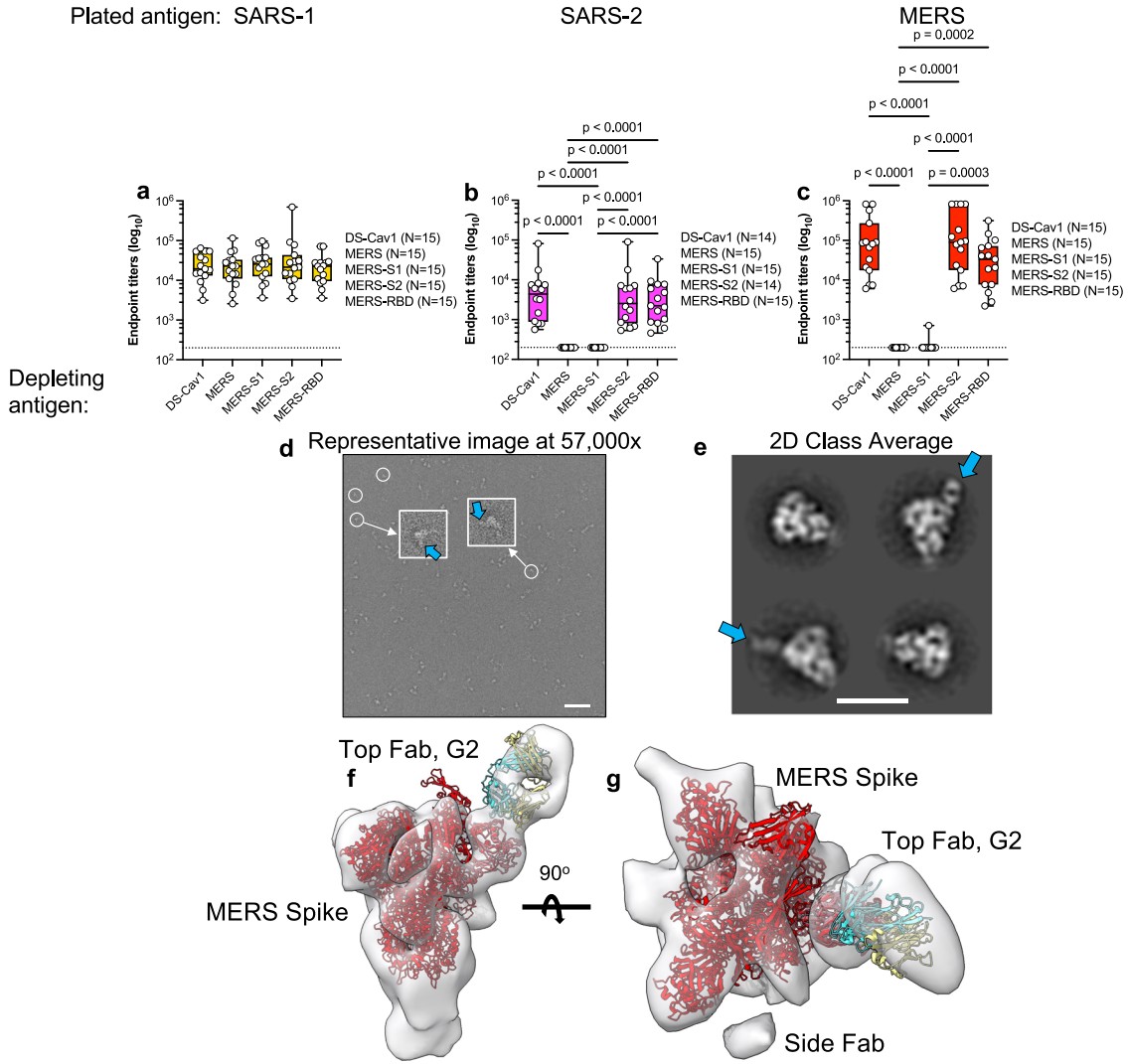

**Fig. 3 | SARS_dn5 elicits two distinct antibody populations targeting the S1 domain of MERS_S-2P. a–c** To elucidate cross-reactive domain specificity, SARS-1_dn5 sera was depleted with MERS_S-2P and its domains, S1, SS, and RBD, then screened for residual binding to **a** SARS-1_S-2P, **b** SARS-2_S-2P, and **c** MERS_S-2P. **d–g** Mice immunized with SARS-1_dn5 were terminally bled and serum was pooled, IgG-purified and digested to Fabs. Immunocomplexes of SARS-1_dn5-elicited Fabs bound to MERS_S-2P were imaged with negative stain EM. **d, e** Squares are magnified views of Fabs bound within the image. Scale bars correspond to 100 nm

(representative image) and 20 nm (2D classes). Arrows point to Fabs bound to the top or side of MERS_S-2P. **f, g** 3D map reconstruction was generated from NSEM and overlayed with structures of MERS_S-2P and MERS-specific mAb G2. **a–c** Boxes and horizontal bars denote the IQR and medians, respectively. Whisker endpoints are equal to the minimum and maximum values. Circles denote each individual animal. Statistical analysis was performed using the nonparametric Kruskal–Wallis test with Dunn's multiple comparisons. **P < 0.01, ***P < 0.001, ****P < 0.0001. EMPEM was performed once.

elicited levels of protection against MERS-CoV challenge comparable to MERS_I53_dn5.

## Discussion

Here, we adapt a previously described computationally designed protein nanoparticle platform[14] to co-display CoV S-2P antigens. Our results demonstrate that the breadth and quality of cross-reactive antibodies elicited by monotypic nanoparticle display of hCoV S-2P is extensive. However, mosaic display of hCoV S-2P proteins on the I53_dn5 platform can achieve similar potency of antibody responses compared to their monotypic counterparts, despite the lower molarity of S-2P immunogens from each individual hCoV, without sacrificing breadth. Others have previously shown that in the context of influenza, mosaic display of diverse hemagglutinins (HA) on I53_dn5 elicits both potent head-directed antibody responses as well as cross-reactive responses against conserved epitopes on the stem and achieved HA-stem-specific antibody-mediated protection against highly divergent

heterosubtypic strains of influenza[22]. In both that work and our present study, the two-component nature of the computationally designed nanoparticle scaffold facilitated co-display through simple mixing of independently purified subunits in vitro[18,22,23].

There was a notable lack of cross-reactive antibody responses directed to the more conserved S2 subdomain when using SARS-1_I53_dn5 as an immunogen. However, SARS-1_I53_dn5 still induced cross-reactive binding and neutralizing responses to the S1 domain of MERS S. We also found a reduction in cross-reactive binding titers elicited by SARS-1_I53_dn5 against SARS-2_S-2P compared to those against MERS_S-2P, despite higher overall conservation and sequence homology between SARS-1_S-2P and SARS-2_S-2P. The regions with the greatest sequence conservation are concentrated in the S2 subunit, where there is approximately 90% amino acid identity between the SARS-1-CoV and SARS-CoV-2 spikes[24]. However, our data suggest that the display of CoV_S-2P on the I53_dn5 nanoparticle initially restricts BCR access to the S2, biasing toward the selection of B cells that

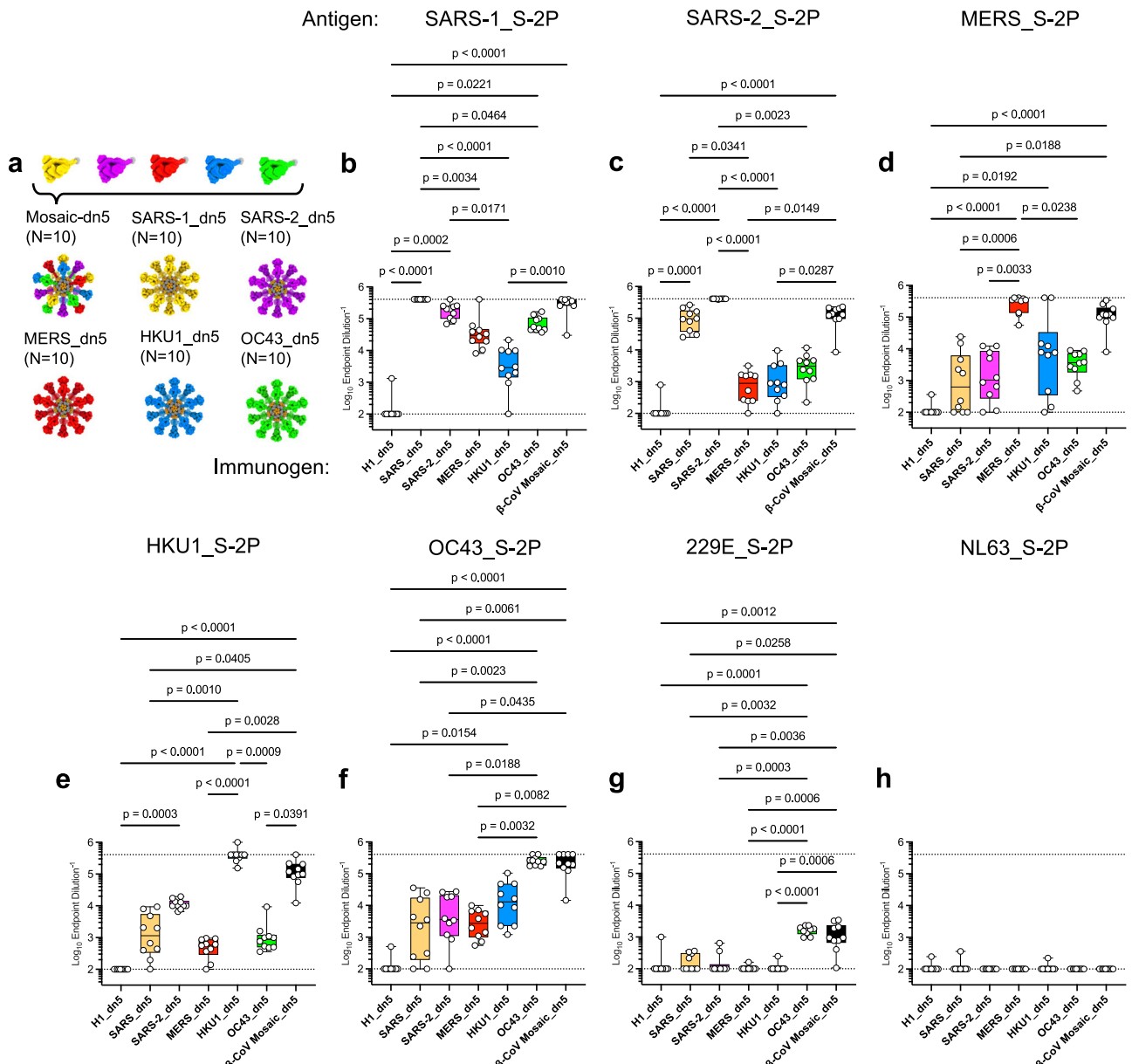

**Fig. 4 | b-mosaic particles elicit broad antibody responses. a** Groups of 10 female BALB/cJ mice were immunized twice with β-CoV mosaic_I53_dn5, MERS_I53_dn5, SARS-1_I53_dn5, SARS-2_I53_dn5, HKU1_I53_dn5, or OC43_I53_dn5 to compare antibody responses elicited from co-display and monotypic display of each spike. Control mice were immunized with H1_I53_dn5. Mice were bled at week 5 for serology. **b**–**h** Sera were screened by ELISA for IgG binding to each strain. Boxes and horizontal bars denote the IQR and medians, respectively. Whisker endpoints are equal to the minimum and maximum values. Circles denote each individual animal. Immune profiling in BALB/cJ performed once.

recognize shared epitopes on the S1 domain. This could be explained by the larger size of S (relative to HA) occupying more space on the nanoparticle scaffold and crowding out BCR access to S2[25], an effect that was confirmed in previous studies of two-component nanoparticle immunogens displaying HIV-1 Env trimers[26,27]. Given the structural and mechanical homology between prefusion coronavirus spikes, we suspect that the apparent S2-occlusion/S1-targeting may also occur with other CoV-S_dn5 formulations, potentially eliciting potent S1-directed cross-neutralization in other coronaviruses as well— particularly in the β-mosaic that may exploit vulnerable S1 epitopes. However, we propose that it may be possible to improve access to S2 epitopes with nanoparticle scaffolds that appropriately space the displayed spike antigens.

Notably, each monotypic CoV_S-2P assembled on I53_dn5 elicited a distinct cross-reactive antibody profile to MERS_, SARS-1_, SARS-2_,

OC43_, 229E_, NL63_, and HKU1_S-2P. Qualitative differences between antibody responses elicited by I53_dn5 immunogens were further exemplified by the observation that while SARS-1_I53_dn5 and SARS-2_I53_dn5 elicited a similar cross-reactive profile and comparable MERS_S-2P-reactive antibody binding titers (Fig. 4b, h and Supplementary Fig. 7b–h), SARS-1_I53_dn5 protected mice from lethal MERS challenge while SARS-2_I53_dn5 did not (Fig. 5a). Additionally, the potency of cross-reactivity appeared to be unidirectional as immunization with MERS_I53_dn5 elicited SARS-1-binding, but non-neutralizing antibodies. Such instances may hint at the targeting of partially conserved epitopes of vulnerability as opposed to the protection conferred merely through antibody quantity or affinity. Depletion data showed that the cross-reactive antibodies elicited by SARS-1_I53_dn5 to SARS-2_ and MERS_S-2P must bind the MERS S1 domain and likely bind shared epitopes.

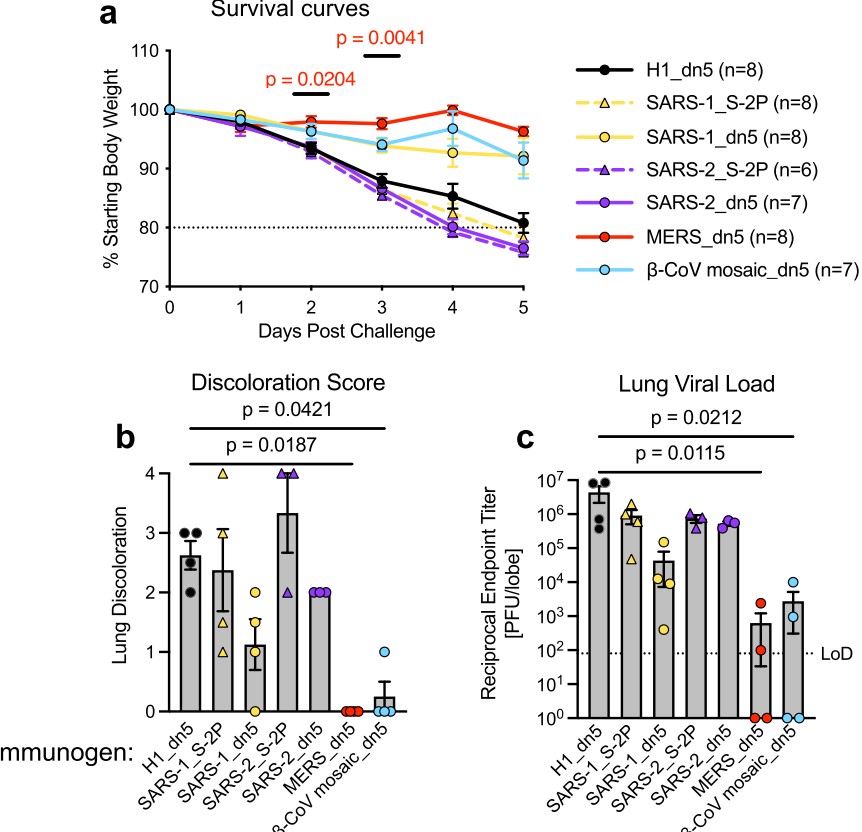

**Fig. 5 | β-CoV mosaic particles protect against lethal MERS-CoV challenge.** 288/330[+/+] mice were immunized twice with 10 µg of the specified nanoparticle or soluble trimer. Control mice were immunized with H1_dn5. 4 weeks post-boost, mice were challenged with a lethal dose, $5 \times 10^5$ plaque-forming units (p.f.u.) of maM35c4 MERS-CoV. **a** Following challenge, mice were monitored for weight loss. **b**, **c** 5 days post challenge, **b** lung discoloration (scored as: 0 = no discoloration, 4 = severe discoloration in all lobes) and **c** lung viral titers were assessed. All groups were compared with H1_dn5 control mice by Kruskal–Wallis analysis of variance (ANOVA) with Dunn's multiple comparisons test; in (**a**), the comparison was made at each day post-challenge. *$P < 0.05$, **$P < 0.01$. Data depict mean ± s.d. in (**a**, **b**) or GMT ± geometric s.d. in (**c**). In (**c**), the dotted line represents the assay limit of detection. Challenge experiments were performed once.

EMPEM of SARS-1_I53_dn5-derived IgG Fabs bound to MERS_S-2P showed at least two cross-binding antibody species. The most easily identified appears to be a G2-like Ab that binds an epitope on the apex of the NTD[20]. Notably, this epitope and Ab binding angle is also analogous to a conserved and previously described SARS-2 antigenic supersite[28]. The second cross-reactive species is visible by 2D class average EMPEM images, which revealed a fab binding to the side of MERS_S-2P. The 3D map reconstruction was unable to reveal a clear binding site or angle of this Fab, indicating flexibility or dynamic motion of the region to which the Fab is bound. Diminished antibody binding from depleted sera further suggests that this species may bind motile MERS_RBD. While these SARS-1_I53_dn5-elicited antibodies failed to neutralize the SARS-2 pseudotyped virus, they did bind SARS-2_S-2P. This binding was abrogated when depleted with MERS S1, suggesting that these antibodies likely bind shared epitopes. However, MERS S1 depletion did not appear to impact binding to HKU1_S-2P, suggesting that any epitopes responsible for cross-reactivity between SARS-1_, SARS-2_S-2P, and MERS_S-2P are not shared with hCoV-HKU1 and are bound by different antibody populations (Supplementary Fig. 2). These findings, in combination with the unidirectional cross-neutralization of SARS-1_I53_dn5 to MERS_CoV (but not MERS_I53_dn5 to SARS-CoV-1), hints at distinct structural requirements for epitope recognition by naïve B cells and the subsequent induction of cross-neutralizing antibodies.

Other studies have reported intrasubgroup cross-reactivity through natural infection[7] or via nanoparticle-based immunogens using mosaic co-display of RBDs from diverse SARS-like CoV S proteins[9,10]. We chose to display the full-length S ectodomain because it comprises the full complement of epitopes exposed on the spike on the viral surface, including the highly conserved S2 fusion machinery, and therefore provides a greater chance of identifying cross-reactive epitopes between diverse CoVs. However, the finding that S2-specific antibodies were not induced suggests that BCR access may have been limited to the apex of S. New designs with fewer S-2P displayed or larger nanoparticles to change the geometry of antigen presentation and improve access to S2 could increase the level cross-reactivity induced. Additionally, employing an S2-only display of diverse hCoV spikes and employing our mosaic approach may increase recognition of conserved epitopes on this subdomain. Nonetheless, our results confirm the presence of conserved protective epitopes between divergent subgroups on the NTD and potentially either RBD or other subdomains that can be targeted by next-generation immunogens.

The β-CoV_I53_dn5 mosaic nanoparticle, which has a relatively low valency of MERS_S-2P trimers, induced protective immunity against MERS-CoV lethal challenge in mice, comparable to protection elicited by monotypic MERS_I53_dn5 (Fig. 5). Further studies will need to evaluate the extent to which this mosaic nanoparticle protects against other CoV challenges and establish the minimum valency of S-2P content needed to confer protection in both homotypic and heterotypic challenge models. Future research will also be needed to map additional cross-reactive epitopes and B cell responses following CoV_I53_dn5 immunization, and additional experimentation will be required to define the optimal spacing and composition of S-2P antigens to improve the breadth and potency of cross-reactive responses.

Follow-up experiments should evaluate the range of protective immune responses against a wider variety of β-CoVs and more distantly related α-CoVs. Additionally, distinct sensitivities to SARS-CoV-2 variants between humans and mice have recently been shown[29], and this platform merits evaluation in additional animal models.

Typically, as breadth increases, neutralizing activity decreases, thus highlighting the importance of understanding orthogonal antibody functions. Studies from both vaccinated and convalescent individuals have shown that neutralizing activity is highly predictive of protection from SARS-CoV-2 infection and that even low levels of neutralization are sufficient for protection[30]. Therefore, more experiments should be conducted to determine the mosaic nanoparticle design that confers maximum breadth while still retaining protection and potency.

Ongoing SARS-CoV-2 circulation highlights the need for protection against pre-emergent zoonotic CoV threats in addition to continually emerging SARS-CoV-2 VOC. The mosaic display of S-2P antigens on nanoparticles may serve as the basis for developing a pan-CoV vaccine.

## Methods

### Expression and purification of antigens and immunogens
Soluble CoV_S-2P trimers with foldon, as well as CoV_S-2P_I53_dn5b nanoparticle components, were expressed in Expi293F expression system (Thermo Fisher Scientific, Cat#: A14527) with Expifectamine transfection reagent according to the manufacturer's protocol. For SARS-CoV-2, the spike sequence derived from isolate Wuhan-Hu-1 was used. Soluble CoV_S-2P was harvested by centrifugation at 3700–4000×g, and clarified supernatant was filtered through 0.22–0.45 μm vacuum filters. Protein was purified by tangential flow filtration, followed by strep tag purification and 3C cleavage of tags. CoV_S-2P_I53_dn5b nanoparticle components were expressed in Expi293 cells (Thermo Fisher Scientific) and purified by nickel resin, size exclusion chromatography using a Superdex 200 increase, and Superose 6 increase (GE Healthcare) was used to purify assemble particles. Briefly, the salt and pH of cleared transfection supernatants were adjusted with Tris-HCl (pH 8.0) buffer at a final concentration of 50 mM Tris-HCl, ~50 mM NaCl. Nickel resin was washed with several bed volumes of either 1x PBS or water 3 times and incubated with supernatant overnight at 4 °C or 2 h at room temperature. Resin-supernatant mixture was loaded onto a column. The column was washed twice with 5 times bed volume of wash buffer (500 mM NaCl, 50 mM Tris-HCl, 5 mM imidazole) and eluted twice with 5 times bed volume elution buffer (500 mM NaCl 50 mM Tris-HCL, 300 mM imidazole). Eluates were concentrated and filtered through 0.22um and purified by size exclusion chromatography using a Superose 6 increase (GE Healthcare). Fractions containing CoV_S-2P_I53_dn5b components were pooled and concentrated for particle assemblies. In vitro assembly of particles was performed by addition and vigorous mixing by pipette of equimolar amounts of I53_dn5a and I53_dn5b components for 20 s at room temperature and left at rest to assemble for 30 min. Assemblies were then purified by size exclusion using Superose 6 increase (GE Healthcare), and fractions were collected.

### Antibody expression and purification
Paired VH and VL in a 1:1 ratio were co-transfected transiently into EXPI293 cells. The supernatant was harvested 6 days post-transfection, and IgGs were purified with Protein A agarose (Thermo Fisher Scientific). IgGs were eluted with 100 mM glycine, pH 3 into 1/10th volume 1 M Tris-HCl pH 8.0. IgGs were then buffer exchanged into PBS pH 7.4. Fabs were generated by digesting the IgGs with HRV 3C protease at 4C. Fc was removed by passing digests over fresh Protein A agarose, leaving the Fab in the flowthrough, which was further purified by SEC using a Superdex 200 increase 10/300 column (GE Healthcare) in PBS buffer, pH 7.4. Antibodies are validated for binding to specific and non-specific antigens by antigenicity ELISA as described below prior to use in experiments.

### Antigenicity testing by enzyme-linked immunosorbent assay (ELISA)
Enzyme-linked immunosorbent assay (ELISA) was used to measure antibody binding to each immunogen. In this, 96-well ELISA plates were coated with 1 μg/mL nanoparticles or CoV_S-2P. Plates were incubated at 4 °C overnight and blocked with PBS containing 5% skim milk at room temperature for 1 h. Monoclonal antibodies were serially diluted in four-fold steps from a starting concentration of 5 μg/mL, then added to the plates and incubated at room temperature for 1 h. Horseradish peroxidase (HRP)-conjugated anti-human IgG (Southern Biotech, Cat 2040-05) was added in 1:5000 dilution in milk block and incubated at 37 °C for 1 h, followed by 3,3′,5′,5- Tetramethylbenzidine (TMB; Sigma-Aldrich, St. Louis, MO) HRP substrate, and the signal that developed after the addition of 1 M sulfuric acid was measured by absorbance at 450 nm.

### Negative stain electron microscopy
Proteins were diluted with buffer containing 10 mM HEPES, pH 7.0, and 150 mM NaCl to a concentration of approximately 0.05 mg/ml and adsorbed to a freshly glow-discharged carbon-coated grid. The grid was washed three times with the same buffer and stained with 0.7% uranyl formate. Images were collected at a nominal magnification of 57,000 (pixel size: 2.53 Å) using EPU software version 2.12.1.2782REL on a Thermo Fisher Scientific Talos F200C electron microscope equipped with a 4k x 4k Ceta camera or at a nominal magnification of 50,000 (pixel size: 4.4 Å) using SerialEM[31] on an FEI T20 electron microscope equipped with a 2k x 2k Eagle camera. Both microscopes were operated at 200 kV. Particles were picked automatically using EMAN2[32] or in-house written software (YT, unpublished). Reference-free 2D classifications were performed using RELION[33]. To obtain the 3D NS-EM map of the immunocomplex of SARS-1_I53_dn5-derived Fab and MERS_S-2P, micrographs were collected at 0- and 30-degree tilt using the same microscope settings. RELION 3.1 was used to isolate the population of particles representing the immunocomplex using 2D and 3D classification and for the refinement of the final dataset containing 3927 particles. The resolution, determined at the FSC threshold of 0.5, was 26 Å. 3D map reconstruction of overlapped with G2 and MERS-S-2P structures using ChimeraX 1.3 and Pymol v2.5.

### Animal studies
All mouse experiments were carried out in compliance with all pertinent US National Institutes of Health regulations and approval from the Animal Care and Use Committee (ACUC) of the Vaccine Research Center, from the Institutional Animal Care and Use Committee at the University of North Carolina at Chapel Hill to guidelines outlined by the Association for the Assessment and Accreditation of Laboratory Animal Care and the U.S. Department of Agriculture. All infection experiments were done in animal biosafety level 3 (BSL-3) facilities at the University of North Carolina at Chapel Hill. For immunogenicity studies, female BALB/cJ or C57BL/6 mice aged 6–8 weeks (Jackson Laboratory) were used. Per the experimental design schema outlined, mice were inoculated intramuscularly with protein immunogens adjuvanted with SAS in a 1:1 ratio as previously described[15] and bled for serological assays. For challenge studies to evaluate CoV_I53_dn5 vaccines, 16- to 20-week-old male and female 288/330[+/+] mice[21] were immunized, bled, and challenged, as detailed in Fig. 5a. Mice were challenged with $5 \times 10^5$ PFU of a mouse-adapted MERS-CoV EMC derivative, mM35c4[34]. On days 3 and 5 post-challenge, lungs were collected from selected mice to assess viral titers and discoloration using previously published methods[21,34]. Briefly, caudal right lung lobes were harvested for analysis of viral load by plaque assay. Lung lobes were homogenized in 1 mL of PBS and glass beads. Mice in all

immunization studies were immunized with 10 µg of total protein unless otherwise stated. For 288/330+/+ mice infected with MERS-CoV, mice were monitored up to 5 days post-challenge and were euthanized, and lungs were collected to investigate lung discoloration and viral burden. UNC mice were housed on a 12-h light/dark cycle (7am–7pm), temperature between 20 and 23 °C, humidity 30–70% (usually in the 50 s). VRC mice were housed on a 12-h light/dark cycle (6am-6pm), temperature between 22 and 25 °C, humidity 30–70% (usually around 35%). The sample size for animal experiments was determined on the basis of criteria set by institutional ACUC. Experiments were neither randomized nor blinded.

### Immunogenicity testing by enzyme-linked immunosorbent assay (ELISA)

The binding of immunoglobulin G (IgG) levels to screened antigens was examined as follows: 96-well enzyme-linked immunosorbent assay (ELISA) plates were coated with 1 µg/mL of CoV_S-2P antigen. Plates were incubated at 4 °C overnight and blocked with PBS containing 5% skim milk at room temperature for 1 h. Sera fractions from the immunized mice, serially diluted in four-fold steps and competed with the addition of 50 µg/mL of foldon. Foldon-serum dilutions were then added, and the plates were incubated at room temperature for 1 h. Horseradish peroxidase (HRP)-conjugated anti-mouse IgG (Southern Biotech., Birmingham, AL, Cat 1030-05) was added in 1:5000 dilution in milk block and incubated at room temperature for 1 h, followed by 3,3',5',5- Tetramethylbenzidine (TMB; Sigma-Aldrich, St. Louis, MO) HRP substrate, and the yellow color that developed after the addition of 1 M $H_2SO_4$ was measured by absorbance at 450 nm.

### Depletion assay

Magnetic HIS dynabeads (Thermo Fisher Scientific) were used to deplete mouse sera of antibodies with specificity to MERS S domains, S-2P, S1, S2, and RBD, according to the manufacturer's protocol. For this, 5 µL sera was added to molar equivalents of depleting protein (calculated from 50 µg/mL foldon) in 445 µL of PBS and incubated for 1 h at room temperature. Magnetic beads were washed three times in PBS, resuspended, and 50 µL was added to each serum-protein mixture and incubated at room temperature for 30 min. Beads were separated from the solution by magnetic strip, and after 5 min, the supernatant was collected and used for future assays.

### Pseudotyped lentiviral preparation

Pseudotyped lentiviral reporter viruses were produced as previously described (Corbett et al.[35]). Briefly, HEK293T/17 cells (ATCC CRL-11268) were co-transfected with pCMV-ΔR8.2 (lentiviral backbone) and pHR'-CMV-Luc (reporter genome) plasmids along with plasmids encoding desired S protein from Wuhan-Hu-1 strain (GenBank no. MN908947.3) with a p.Asp614Gly mutation (D614G), MERS England1 strain, and SARS-CoV-1 Urbani strain and human transmembrane serine protease 2 (TMPRSS2) by the Fugene6 transfection method (Promega). After overnight incubation, dishes were washed and replenished with fresh medium. Forty-eight hours later, supernatants were harvested, filtered through a 0.45 µm, aliquoted, and frozen at −80 °C until use. Each pseudotyped virus stock was titrated prior to use in neutralization assays.

### MERS-CoV pseudotyped virus neutralization assay

MERS-CoV pseudotyped virus neutralizing activity was tested as previously described[19]. In brief, Huh 7.5 cells were seeded at $10^4$ cells/well in 96-well black/white Isoplates (PerkinElmer) the day before infection. Sera samples were serially diluted (1:40, 4-fold dilutions, 8x) in DMEM (Gibco)+1% penicillin/streptomycin. Dilutions were mixed with a pseudotyped lentivirus expressing the MERS-CoV England1 spike, which was previously titrated to $10^4$ relative luciferase units (RLU) and

incubated for 30 min at RT. Sera-virus mixtures were added, in duplicate, to cells and incubated at 37 °C and 5% $CO_2$ for 2 h. Next, 100 µL of DMEM (Gibco)+10% FBS+1% penicillin/streptomycin (D10) media supplemented with 2 mM glutamine was added and incubated under the same conditions for 72 h. Cells were then lysed, and luciferase substrate (Promega) was added. Luciferase activity was measured in RLU at 570 nm using SpectraMaxL (Molecular Devices). Considering uninfected cells as 100% neutralization and cells transduced with pseudotyped virus alone as 0% neutralization, averages of duplicates were normalized, and sigmoidal curves were generated. 50% neutralizing antibody titers ($ID_{50}$) titers were generated by fitting normalized values to a log(agonist) vs. normalized-response (variable slope) nonlinear regression model in Prism v9 (GraphPad).

### SARS-CoV-2 and SARS-CoV-1 neutralization assay

SARS-CoV-1 and SARS-CoV-2 pseudotyped virus neutralizing activity was assessed similarly to MERS-CoV pseudotyped virus neutralizing activity. Twenty-four hours prior to infection, HEK293T cells stably overexpressing a humanized ACE2 receptor (provided by M. Farzan, Scripps Research Institute) were seeded in 96-well black/white Isoplates (PerkinElmer) at 5000 cells/well and incubated in 37 °C and 5% $CO_2$. Sera samples were diluted identically as for the MERS-CoV pseudotyped virus assay, mixed with appropriately titrated pseudotyped lentivirus reporter expressing either the SARS-CoV-1 Urbani spike or the SARS-CoV-2 Wuhan-1 Spike encoding the D614G mutation, and incubated for 45 min at 37 °C and 5% $CO_2$. Sera-virus mixtures were then added to cells in triplicate and incubated at the above conditions for 72 h. Cells were lysed, and luciferase substrate (Promega) was added. RLU readings and 50% neutralizing antibody titers (ID50) were determined using the same methodology as for the MERS-CoV pseudotyped virus neutralization assay.

### Statistics

In box plots, boxes and horizontal bars denote the interquartile range and medians, which represent data outside the 25th–75th percentile range. Whisker endpoints are equal to the minimum and maximum values. Geometric means or arithmetic means are represented by the heights of bars or symbols, and error bars represent the corresponding s.d. Dotted lines indicate assay limits of detection. Figure legends detail all quantification and statistical analyses, inclusive of animal numbers (*n*), dispersion and precision measures, and statistical tests performed. To compare more than two experimental groups, Kruskal–Wallis ANOVA with Dunn's multiple comparisons tests were applied, assuming a non-Gaussian distribution. Reciprocal endpoint and reciprocal $ID_{50}$ titers were transformed so that all values were on a $log_{10}$ scale prior to statistical analyses (Figs. 2, 3, 5c). Statistical analyses were performed using GraphPad Prism v9.2.0. Significance is denoted by asterisks defined as *$P < 0.05$, **$P < 0.01$, ***$P < 0.001$, ****$P < 0.0001$.

### Reporting summary

Further information on research design is available in the Nature Portfolio Reporting Summary linked to this article.

## Data availability

Source data supporting the findings of this study are available within this paper and its supplementary files. Source data has also been deposited to Figshare and can be accessed here: https://doi.org/10.6084/m9.figshare.24052845. No deposited structures were generated in this study. 3D map overlays in Fig. 3f, g utilized previously published/deposited structures: PDB: 5X5F and PDB: 6PXH.

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

## Acknowledgements

We would like to thank Tracy Ruckwardt, Emily Phung, Anthony DiPiazza and Brian Fisher for critical discussions regarding cross-reactivity and attempts to sort cross-reactive B cell clones. We thank Judy Stein and Monique Young for technology transfer and administrative support, respectively. We thank members of the NIH NIAID VRC Translational Research Program for technical assistance with mouse experiments. This work was supported by the Intramural Research Program of the VRC and the Division of Intramural Research, NIAID, NIH (B.S.G.). MERS mouse challenge studies were funded under NIH Contract HHSN272201700036 Task Order No. 75N93019F00132 Requisition No. 5494549 (to R.S.B.). K.S.C.'s research fellowship was partially funded by the Undergraduate Scholarship Program, Office of Intramural Training and Education, Office of the Director, NIH. The publication was supported by Springer Nature Inspiring Women in Science Award— Achievement Award (to K.S.C.). This work was supported in part with federal funds from the Frederick National Laboratory for Cancer Research, NIH, under Contract HHSN261200800001 (Y.T.), the Bill & Melinda Gates Foundation (OPP1156262 to D.B. and N.P.K.), the Audacious Project at the Institute for Protein Design (D.B. and N.P.K.), the Division of Extramural Research of the National Institutes of Allergy and Infectious Diseases through P01AI167966 (N.P.K and R.S.B.) and K.S.C.'s start-up funds from Harvard T.H. Chan School of Public Health.

## Author contributions

G.B.H. and D.E. engineered nanoparticles using I53_dn5 sequences from G.U. and S_2P sequences from N.W. and D.W. G.B.H., O.M.A., S.R.L., A.W. and E.J.F. designed and conducted animal experiments. G.B.H., O.M.A., C.T.Z. and A.P.W. performed ELISAs. O.M.A., C.T.Z., A.P.W. and S.B.B. performed pseudovirus neutralization assays. G.B.H. and D.E. purified IgG, digested Fabs, and assembled immune complexes. Y.T. performed

NSEM and 3D map analysis. C.P. performed bioinformatics. G.B.H., O.M.A. and K.S.C. interpreted the data. G.B.H., K.S.C. and B.S.G. conceptualized the study. R.S.B., J.S.M., D.B., K.S.C., M.K., N.P.K. and B.S.G. supervised, administrated, and raised funding for the project. G.B.H. wrote the manuscript with input from all the authors. A.W., C.P., O.M.A., K.S.C. and B.S.G. revised and polished the manuscript.

## Competing interests

B.S.G., J.S.M., M.K., N.W., and K.S.C. are inventors on US Patent no. 10,960,070 entitled 'Prefusion Coronavirus Spike Proteins and Their Use.' K.S.C., O.M.A., G.B.H., N.W., D.W., J.S.M., and B.S.G. are inventors on US Patent Application No. 62/972,886 entitled "2019-nCoV Vaccine". D.E., G.U., N.P.K., B.S.G., K.S.C., M.K., and G.B.H. are inventors on US patent application No. 63/022,214 entitled "Nanoparticle vaccines for coronaviruses". N.P.K. is a co-founder, shareholder, paid consultant, and chair of the scientific advisory board of Icosavax, Inc. The King lab has received unrelated sponsored research agreements from Pfizer and GSK. All other authors declare no competing interests. All other authors declare no competing interests.

## Additional information

[1]Vaccine Research Center, National Institute of Allergy and Infectious Diseases, National Institutes of Health, Bethesda, MD, USA. [2]Institute for Protein Design, University of Washington School of Medicine, Seattle, WA, USA. [3]Department of Immunology, University of Washington School of Medicine, Seattle, WA, USA. [4]Case Western Reserve University, Cleveland, OH, USA. [5]Department of Molecular Microbiology and Immunology, Johns Hopkins University Bloomberg School of Public Health, Baltimore, MD, USA. [6]Department of Biochemistry, University of Washington School of Medicine, Seattle, WA, USA. [7]Vaccine Research Center Electron Microscopy Unit, Leidos Biomedical Research Inc., Frederick National Laboratory for Cancer Research, Frederick, MD, USA. [8]Department of Epidemiology, Gillings School of Global Public Health, University of North Carolina at Chapel Hill, Chapel Hill, NC, USA. [9]Department of Immunology and Infectious Diseases, Harvard T.H. Chan School of Public Health, Boston, MA, USA. [10]Department of Microbiology and Immunology, School of Medicine, University of North Carolina at Chapel Hill, Chapel Hill, NC, USA. [11]College of Natural Sciences, University of Texas at Austin, Austin, USA. [12]Howard Hughes Medical Institute, University of Washington, Seattle, WA, USA. [13]These authors contributed equally: Geoffrey B. Hutchinson, Olubukola M. Abiona. ✉e-mail: bgraham@msm.edu; kizzmekia_corbett@hsph.harvard.edu

