## [Peer Review File · Nature Communications]

Nanoparticle display of prefusion coronavirus spike elicits S1-focused cross-reactive antibody response against diverse coronavirus subgeneraREVIEWER COMMENTS

Reviewer #1 (Remarks to the Author):

Summary

This article provides an investigation of vaccine candidates targeting beta-coronaviruses. Comparative studies look at 2P spike trimer protein-immunized versus engineered 2P trimer-displaying nanoparticles. Looking at several individual coronavirus trimers, single immunogen nanoparticle immunogens appeared to offer a small benefit in terms of relative quality of induced antibodies. However, these do not improve the robustness or breadth of the antibody response. The authors move to a deeper comparison of nanoparticle displayed trimers including a mosaic version incorporating all other trimers utilized into a single vaccine immunogen. Interestingly, this mosaic approach induces a broader antibody binding response, and one capable of protection in a MERS challenge model.

Major concerns:

Given the importance of SARS-CoV-2 VOC and breakthrough from existing vaccine responses by VOCs, it is of urgent necessity to have new technology to make vaccines or monoclonal therapy. This study connects to other studies already in the literature attempting to use nanoparticles as part of a broad coronavirus vaccination strategy. While others focus more specifically on sarbecoviruses, or only on SARS-CoV-2 variants, the authors here are attempting to identify suitable vaccine approaches for the larger betacoronavirus family. This is indeed a worthwhile approach but are a number of limitations to the study.

1. The immune responses were only biased to S1 region and the authors hypothesis that the S2 region was sterically restricted from BCR access on the nanoparticles surface. The authors hypothesized that future work would attempt to alter immunogen spacing to improve S2 access. This seems critical as S2 has been shown to be a key region for eliciting cross-neutralizing antibodies (Ex. Ng, K.W., et al. (2022). SARS-CoV-2 S2-targeted vaccination elicits broadly neutralizing antibodies. *Sci Transl Med* 14, eabn3715. 10.1126/scitranslmed.abn3715). As the authors discuss, the focusing of the immune system on S1 region is something that needs resolved for a more promising vaccine candidate. While the manuscript is interesting, it is somewhat incremental and mostly relevant to a more limited scientific audience. This reviewer would expect either resolution of this limitation (ex. the authors proposed increased spacing nanoparticle), or combination of the studied nanoparticles with those capable of also eliciting antibodies to other conserved regions. Alternatively, the authors could show a broader in vivo protection profile for the mosaic particles. Regardless, in its current format it is more appropriate for a more narrow audience of investigators focused on interactive coronavirus vaccine designs.

2. Figure 2.e. It is surprising to see that there was no SARS-2 pseudotyped virus neutralization from SARS1-S2P or SARS-dn5 nanoparticles. As, the authors claim in the discussion that there are conserved epitopes between SARS and MERS NTD and hence see protection. But SARS1 and SARS2 have many more conserved sites or epitopes compared to MERS. Can authors explain these results?

3. Fig 2g-h. It is interesting that nanoparticle displayed SARS1 antigen has a higher neutralization to titer ratio. However, if in the end a vaccine induces exceptionally high-quality antibodies, but at a low titer, it still is likely to be less efficacious than a moderate. Can the authors comment on this potential limitation?

4. The most exciting finding in this manuscript is the potential protection afforded by the Beta-mosaic nanoparticles. These were shown to be cross-reactive to other betacoronaviruses and protective against an in vivo MERS challenge. However, some idea of neutralizing potency of the induced antibodies from this approach is not provided. Mice were immunized and characterized for binding breadth, but not neutralization. As the monotypic trimer-displaying nanoparticles showed pseudotype viral neutralization, it is presumed that the mosaic particle elicits neutralizing antibodies as part of the induced protection. Showing these data would allow the reader to more completely assess the potential for cross-protection of other betacoronaviruses.

Minor comments:

1. I recommend authors use pseudotyped virus rather than pseudovirus.

2. Figure 5. Please indicate p value or * in the figure, although the authors seems to mention it in the legend.

Reviewer #2 (Remarks to the Author):

The goal of this manuscript is to generate high-valency nanoparticles that are displaying single spike species or mosaic display of diverse betaCoV spike proteins to investigate how these nanoparticles improve immunogenicity, neutralizing breadth, and protection. The manuscript is well written and fairly easy to follow, with the exception of all of the different antigens used. The study is very interesting and provides rationale for how immunity against particular CoVs can cross-react with distant related CoVs. My concerns are mostly minor.

Major Concerns

1. Figure 2a-c and supplementary 3 a-c – It's not clear which isotype is being picked up here, although methods suggest it is IgG (all subclasses?). As SARS-1-S-2P induced higher titers, again not sure isotype, but with lower potency as those induced by is it possible SARS-1_153_dn5 promoted better germinal center responses, and ultimately more IgG/higher affinity antibodies relative to mice that received the monovalent form?
2. Expansion of methods section to include information of how much of the monovalent spike were used would be helpful. As is, it isn't clear if equimolar concentrations of spike were given. Also at like 414-415, the authors mention they used previously published methods but do not cite these methods.
3. For figure 4 – are these antibodies actually cross-reactive or are they subtype-specific antibodies against all the different betaCoVs? The mosaic antigen and the OC43 antigen both have comparable titers against 229E, so the antibodies induced by the mosaic antigen may just be the OC43 cross-reactive antibodies or could be cumulative from other betaCoVs. Sera depletion studies would help clarify this point.
4. Map of the constructs would be helpful, notably if any flexible linker was included in the construct.
5. A table with all of the different antigens used and a brief description would be very helpful for keeping everything straight.

Minor Concerns

1. Line 14 – at this point it is not clear what SARS-I53_dn5 is.
2. It would be helpful to clarify that I53_dn5 is an icosahedron in the text.
3. Line 83 – please describe what the Sigma adjuvant system is
4. Line 158 – what is MERS_SS? Line 162 suggests it S2 domain.
5. Line 173 – Can you overlay the top Fab with G2?

Reviewer #3 (Remarks to the Author):

In this manuscript, the authors developed multivalent protein nanoparticles displaying CoV_S-2P trimers to elicit neutralizing especially cross-neutralizing antibodies against distant group β -coronaviruses. They

also designed the mosaic nanoparticles co-displaying distinct CoV spike as spectrum vaccine and verified it in MERS-CoV challenged mice. Based on their results, the immunogenicity profile of spike displayed on nanoparticles was different from spike itself, possibly caused by the cover-up of S2 domain. This phenomenon is interesting, and the author tried to explain it through serum antibody depletion experiments and negative stain EM. However, some issues still need to be addressed to emphasize the advantages of nanoparticle display of viral antigen as a vaccine.

1. The cross-reactive was only observed in SARS nanoparticles against MERS-CoV and not in SARS-CoV-2 nanoparticles. Does this cross-reactivity only occur in specific settings? Did the author compare the cross-reactivity of nanoparticles and the protein itself using other viral spikes? Like in figure 4, how about the performance of these CoV_dn5 compared with CoV_S-2P?
2. Although SARS-1_dn5 induced a group of MERS-CoV cross-reactive antibodies which were not shown in SARS-1_S-2P group, the nanoparticle seems didn't show a better ability to elicit corresponding antibodies to SARS-CoV than the SARS spike itself from data presented in figure 2. These make the superiority of nanoparticle display antigens indistinctive. Did the author compare the long-term protection efficacy of these vaccines? Or did the author analyze the cell-mediated immune response induced by these two kinds of vaccines? Will they induce different T cell responses? And in the MERS-CoV challenged animal protection experiment in figure 5, did the author perform a group of MERS_S-2P vaccine to highlight the better in vivo efficacy of MERS-CoV nanoparticles? The author may need to try to further address the advantages of nanoparticle display vaccines.
3. Did the author look into the safety of this large molecular-weight protein nanoparticle? For example, hepatotoxicity.
4. The assembling of 20 copies trimer seems to bury the base domain of S2, does that account for the lost potency compared with spike protein? If that, is this nanoparticle better than the recombinant S1 or RBD subunit vaccine?

Minors:

1. Figure legend of figure 1 is not consistent with the figure
2. Figure 2e, x axis, "SARS_dn5" should be changed to "SARS-1_dn5", and also in figure 4h.
3. Figure 5b,c, figure legend mentioned the significance analysis as " $*P < 0.05$, $**P < 0.01$ ", but "*" was not marked in the figure.

Reviewer 1:

Major concerns: Given the importance of SARS-CoV-2 VOC and breakthrough from existing vaccine responses by VOCs, it is of urgent necessity to have new technology to make vaccines or monoclonal therapy. This study connects to other studies already in the literature attempting to use nanoparticles as part of a broad coronavirus vaccination strategy. While others focus more specifically on sarbecoviruses, or only on SARS-CoV-2 variants, the authors here are attempting to identify suitable vaccine approaches for the larger betacoronavirus family. This is indeed a worthwhile approach but are a number of limitations to the study.

1. The immune responses were only biased to S1 region and the authors hypothesis that the S2 region was sterically restricted from BCR access on the nanoparticles surface. The authors hypothesized that future work would attempt to alter immunogen spacing to improve S2 access. This seems critical as S2 has be shown to be a key region for eliciting cross-neutralizing antibodies (Ex. Ng, K.W., et. al. (2022). SARS-CoV-2 S2-targeted vaccination elicits broadly neutralizing antibodies. *Sci Transl Med* 14, eabn3715. 10.1126/scitranslmed.abn3715). As the authors discuss, the focusing of the immune system on S1 region is something that needs resolved for a more promising vaccine candidate. While the manuscript is interesting, it is somewhat incremental and mostly relevant to a more limited scientific audience. This reviewer would expect either resolution of this limitation (ex. the authors proposed increased spacing nanoparticle), or combination of the studied nanoparticles with those capable of also eliciting antibodies to other conserved regions. Alternatively, the authors could show a broader in vivo protection profile for the mosaic particles. Regardless, in it's current format it is more appropriate for a more narrow audience of investigators focused on interactive coronavirus vaccine designs.

We thank the reviewer for their thoughtful and critical feedback. Current literature in the field of antigen spacing, particularly in the context of nanoparticles, is a complex area of molecular immune-engineering that is outside the scope of this manuscript. However, we currently have multiple studies underway to better investigate how spatial and geometric positioning of viral antigens on a nanoparticle can influence immunogenicity. Understanding that S2 has been shown to be a key region for eliciting cross-neutralizing antibodies, we also engineered a MERS stabilized stem (S2) antigen (*Cell Reports*, 2021) that we are currently displaying on nanoparticles. As the reviewer highlights, most nanoparticle vaccine candidates focus heavily on improving sarbecovirus breadth. Here, we chose to highlight MERS-CoV protection to solidify the point that our mosaic nanoparticles are useful for pandemic preparedness as MERS-CoV is antigenically distinct from the sarbecoviruses.

Figure 2.e. It is surprising to see that there was no SARS-2 pseudotyped virus neutralization from SARS1-S2P or SARS-dn5 nanoparticles. As, the authors claim in the discussion that there are conserved epitopes between SARS and MERS NTD and hence see protection. But SARS1 and SARS2 have many more conserved sites or epitopes compared to MERS. Can authors explain these results?

While we think that there is a conserved epitope, it is possible that there are subtle positional differences in the location of the epitope between SARS-1 and SARS-2 spikes that could alter an antibody's binding angle such that it would no longer inhibit receptor binding, and thus does not neutralize SARS-2 pseudotyped virus. While beyond the scope of this manuscript, we currently have plans to isolate monoclonal antibodies from dn5 immunized mice to better elucidate binding angle and exact amino acids that comprise the epitope(s).

2. Fig 2g-h. It is interesting that nanoparticle displayed SARS1 antigen has a higher neutralization to titer ratio. However, if in the end a vaccine induces exceptionally high-quality antibodies, but at a low titer, it still is likely to be less efficacious than a moderate. Can the authors comment on this potential limitation?

The results of binding and neutralization data are indicative of a potentially neutralizing epitope of vulnerability that's novel in its conservation between highly divergent strains of coronavirus. Additionally, it is important to note instances of high binding antibody titers with poor neutralization in which binding does not necessarily improve protection (as may be seen with RSV). As such, identifying sites of potent neutralization is critical to rational vaccine design, even if initial binding titers are relatively low. While this is demonstrated in the context of vaccination, this finding is also critical for passive immunization strategies. We point to therapeutic monoclonal antibodies--while multiple sites of vulnerability are ideal, relatively few potentially neutralizing abs can confer protection. As well, vaccines strategies that combine multiple conserved and potent targets will be most effective at protecting against diverse infections. Our future work will employ strategies to enhance binding to the epitope identified here as well as other conserved sites of vulnerability. Accumulation of these targets will enhance overall binding while retaining neutralizing potency and likely durability.

3. The most exciting finding in this manuscript is the potential protection afforded by the Beta mosaic nanoparticles. These were shown to be cross-reactive to other betacoronaviruses and protective against an in vivo MERS challenge. However, some idea of neutralizing potency of the induced antibodies from this approach is not provided. Mice were immunized and characterized for binding breadth, but not neutralization. As the monotypic trimer-displaying nanoparticles showed pseudotype viral neutralization, it is presumed that the mosaic particle elicits neutralizing antibodies as part of the induced protection. Showing these data would allow the reader to more completely assess the potential for cross-protection of other betacoronaviruses.

This is a valid point. Unfortunately, at the time of the experiments we focused on the binding profiles due to the lack of pseudotyped virus neutralization assays against all of the beta-CoVs; this allowed for us to show a complete picture of breadth. Binding antibodies, and not just neutralization, has been shown to be the correlate of protection following SARS-CoV-2 vaccination so while neutralization potency is important, we felt binding titers provided a more wholistic picture of the antibodies elicited. Unfortunately, the serum from these studies has been used in multiple immunoassays, as shown so we cannot complete follow-up neutralization assays.

Minor comments:

1. I recommend authors use pseudotyped virus rather than pseudovirus. – We have changed “pseudovirus” to “pseudotyped virus” throughout the manuscript.
2. Figure 5. Please indicate p value or * in the figure, although the authors seem to mention it in the legend. – We apologize for the inconvenience. It seems our PDF file conversion omitted the p-value notations. This has now been corrected.

Reviewer 2:

(Remarks to the Author): The goal of this manuscript is to generate high-valency nanoparticles that are displaying single spike species or mosaic display of diverse betaCoV spike proteins to investigate how these nanoparticles improve immunogenicity, neutralizing breadth, and protection. The manuscript is well written and fairly easy to follow, with the exception of all of the different antigens used. The study is very interesting and provides rationale for how immunity

against particular CoVs can cross-react with distant related CoVs. My concerns are mostly minor.

Major Concerns:

1. Figure 2a-c and supplementary 3 a-c – It's not clear which isotype is being picked up here, although methods suggest it is IgG (all subclasses?). As SARS-1-S-2P induced higher titers, again not sure isotype, but with lower potency as those induced by is it possible SARS 1_153_dn5 promoted better germinal center responses, and ultimately more IgG/higher affinity antibodies relative to mice that received the monovalent form? We thank the reviewer for highlighting this oversight as we should have specified that these ELISAs are against all subclasses of IgG. We agree with the reviewer and believe that it's highly likely that nanoparticles are stimulating better germinal centers as well as more initial precursor GCBs that may be lower affinity like that shown by Y. Kato, *et. al.*, *Immunity*, 2020. This is likely an important contributing factor to elicitation of cross-reactive antibodies.
2. Expansion of methods section to include information of how much of the monovalent spike were used would be helpful. As is, it isn't clear if equimolar concentrations of spike were given. Also at like 414-415, the authors mention they used previously published methods but do not cite these methods.
Good point, and thank you for catching that. We delivered 10ug of total protein. In the case of nanoparticle vs soluble, given the added weight of the scaffold, there are slightly more molecules of soluble spike than nanoparticle, but we feel this difference is negligible. We have now specified that in the methods for clarity. The citation for the previously published methods has now been added.
3. For figure 4 – are these antibodies actually cross-reactive or are they subtype-specific antibodies against all the different betaCoVs? The mosaic antigen and the OC43 antigen both have comparable titers against 229E, so the antibodies induced by the mosaic antigen may just be the OC43 cross-reactive antibodies or could be cumulative from other betaCoVs. Sera depletion studies would help clarify this point.
For each of the homotypic particles with heterologous antibody binding, these antibodies are truly cross-reactive. In the case of the mosaic nanoparticle, it is likely that antibodies against spikes that are included in the formulation (MERS, SARS, SARS-2, HKU1, and OC43) are a mixture of subtype-specific as well as cross-reactive antibodies. The cross-reactivity from the mosaic is exemplified by the maintained cross-reactivity against 229E. We found it interesting that these 229E-binding antibodies titers appeared comparable to that of the OC43-dn5 particle despite approximate 5-fold lower dose of OC43 spike. Sera depletion studies are a great idea, but we have used all serum from these experiments and believe this type of question would be best answered in a completely different study. With that said, in studies that are beyond the scope of this manuscript, in recently-funded work, we plan to dissect polyclonal serum and isolate monoclonal antibodies from nanoparticle-immunized mice to further delve into breadth of cross-reactivity elicited.
4. Map of the constructs would be helpful, notably if any flexible linker was included in the construct.
A construct list with amino acid sequences and linkers is included in supplemental materials.

5. A table with all of the different antigens used and a brief description would be very helpful for keeping everything straight.
We have now included a table of all the different antigens used and a brief description in the supplement.

Minor Concerns:

1. Line 14 – at this point it is not clear what SARS-I53_dn5 is
Thank you for pointing this out. We have now made it clear.
2. It would be helpful to clarify that I53_dn5 is an icosahedron in the text
Thank you. Please see line 49 where we now clarify that I53_dn5 is icosahedron.
3. Line 83 – please describe what the Sigma adjuvant system is
We now describe what Sigma adjuvant system is in lines 80-82.
4. Line 158 – what is MERS_SS? Line 162 suggests it S2 domain.
We have edited this for clarity.
5. Line 173 – Can you overlay the top Fab with G2?
Yes, please see figure 3 F-G.

Reviewer 3

(Remarks to the Author): In this manuscript, the authors developed multivalent protein nanoparticles displaying CoV_S 2P trimers to elicit neutralizing especially cross-neutralizing antibodies against distant group β - coronaviruses. They also designed the mosaic nanoparticles co-displaying distinct CoV spike as spectrum vaccine and verified it in MERS-CoV challenged mice. Based on their results, the immunogenicity profile of spike displayed on nanoparticles was different from spike itself, possibly caused by the cover-up of S2 domain. This phenomenon is interesting, and the author tried to explain it through serum antibody depletion experiments and negative stain EM. However, some issues still need to be addressed to emphasize the advantages of nanoparticle display of viral antigen as a vaccine

1. The cross-reactive was only observed in SARS nanoparticles against MERS-CoV and not in SARS-CoV-2 nanoparticles. Does this cross-reactivity only occur in specific settings? Did the author compare the cross-reactivity of nanoparticles and the protein itself using other viral spikes? Like in figure 4, how about the performance of these CoV_dn5 compared with CoV_S 2P?

Cross-reactivity does seem to occur in specific settings as there were also some instances of unidirectional cross-reactivity, whereby cross-reactivity is stronger from one spike to another than in the reverse direction. These findings paint a picture of an apparent patchwork of cross-reactivity that might hint at structural changes to the positioning of conserved epitopes that might alter the binding angle of cross-reactive antibodies. For example, in the case of SARS-1, SARS-2, and MERS, it's possible that the epitope is conserved between MERS, SARS, and SARS-2, but mutations in the spike have shifted the position of the epitope. In this hypothetical scenario, antibodies may bind at an angle that interferes with receptor binding from MERS and SARS, but not an angle to interfere with SARS-2. We did not compare the cross-reactivity of nanoparticles and the protein itself using other viral spikes, as such comparisons have been made previously (Y, Kato, *et. al.*, Immunity, 2020., T. Tokatlian, *et. al.*, Science, 2018), and we know based on our own MERS S-2P work (Wang, *et. al.* PNAS, 2017) and SARS-CoV-2 vaccination that cross-reactivity is not particularly broad following S-2P vaccination.

2. Although SARS-1_dn5 induced a group of MERS-CoV cross-reactive antibodies which were not shown in SARS-1_S-2P group, the nanoparticle seems didn't show a better ability to elicit corresponding antibodies to SARS-CoV than the SARS spike itself from data presented in figure 2. These make the superiority of nanoparticle display antigens indistinctive. Did the author compare the long-term protection efficacy of these vaccines? Or did the author analyze the cell-mediated immune response induced by these two kinds of vaccines? Will they induce different T cell responses? And in the MERS-CoV challenged animal protection experiment in figure 5, did the author perform a group of MERS_S-2P vaccine to highlight the better in vivo efficacy of MERS-CoV nanoparticles? The author may need to try to further address the advantages of nanoparticle display vaccines.

It is worth noting that mice with immunized with 10ug of total protein. To prove superiority of the nanoparticle display vs. soluble protein in the context of elicitation of homogenous antibodies, we would have to complete dose titration experiments, which is outside the scope of this manuscript. We were solely focused on display strategies that would improve protective breadth; as such, the distinctive advantage of the nanoparticle over S-2P is precisely the elicitation of MERS-neutralizing antibodies of which SARS S-2P did not elicit. There is already a lot of influential literature in the field looking at distinct advantages conferred through nanoparticle vaccine strategies over soluble antigen. A particularly influential paper is Y. Kato, *et. al.*, *Immunity*, 2020, in which they delve into immunological mechanisms behind how nanoparticle valency alters germinal center B cell selection. While T cell responses were outside the scope of this manuscript that is a topic of interest. S.Hong. *et. al.* *Immunity*, 2018) show that B cells, rather than DCs are the dominant antigen presenting cells in the context of VLPs and are potent activators of CD4 T cells. As such, it's certainly conceivable that there may be differences in how soluble vs nanoparticle immunogens activate T cells if those T cells are being activated by B cells rather than DCs. We used MERS-dn5 as a positive control for the MERS-CoV challenge, where our goal was to show the multivalent nanoparticle protects, not to compare against MERS S-2P.

3. Did the author look into the safety of this large molecular-weight protein nanoparticle? For example, hepatotoxicity.

We feel that hepatotoxicity is not necessary for this particular study as the immunogens are all assembled from recombinant protein and as such is readily degraded in vivo. Additionally, antigen delivered intramuscularly is predominantly drained and trafficked through the lymphatic network and in such relatively small doses that protein burden on the liver would be negligible. Additionally, it is worth noting that this platform adapted to display influenza HA is currently undergoing clinical trials and a similar particle platform I53_50 adapted to display SARS-CoV-2 RBD has been approved and licensed for use in humans. From a safety profile standpoint, protein nanoparticle vaccines are likely comparable to recombinant subunit protein vaccines. Y. Yang, *et. al.* investigates this and shows that: "...The decrease in liver and spleen radioactivity with time implies that the nanoparticles are broken down and cleared. This is an important finding, as it shows that the [self-assembling protein] nanoparticles can be safely used as a vaccine platform without the risk of prolonged side effects." (*J. Nanobiotechnol.*, 2013).

4. The assembling of 20 copies trimer seems to bury the base domain of S2, does that account for the lost potency compared with spike protein? If that, is this nanoparticle better than the recombinant S1 or RBD subunit vaccine?

The burying of S2 on the particle appears to be the most likely explanation for the difference in antibody binding to S-2P. The major appeal to domain subunit vaccines is the ability to focus antibody responses to vaccine target epitopes such as S1 or RBD. However, in theory,

truncating an immunogen, while eliminating less productive antibody responses, could also reduce overall immunogenicity as smaller proteins are often less immunogenic. There is a wealth of literature detailing the distinct advantages of nanoparticle display over subunit vaccines--particularly the ability to induce B cell receptor cross-linking. Additionally, by taking advantage of antigen positioning on the particle that restricts off-target responses, we too can effectively focus antibody responses to S1 without sacrificing the benefits that come with using larger, more complex antigens. We feel that a less-explored benefit of this method of nanoparticle display is the elicitation of more potent breadth of cross-reactive antibodies that are focused to vaccine target domains.

Minors:

1. Figure legend of figure 1 is not consistent with the figure
Thank you for pointing this out. We have changed the figure legend.
2. Figure 2e, x axis, "SARS_dn5" should be changed to "SARS-1_dn5", and also in figure 4h.
Thank you for pointing this out. We have changed this.
3. Figure 5b,c, figure legend mentioned the significance analysis as " $P < 0.05$, $P < 0.01$ ", but "*" was not marked in the figure.
We apologize for the inconvenience. It seems our PDF file conversion omitted the p-value notations. This has now been corrected.